# Gas Plume Target Detection in Multibeam Water Column Image Using Deep Residual Aggregation Structure and Attention Mechanism

Wenguang Chen [1], Xiao Wang [1,*], Binglong Yan [2], Junjie Chen [1], Tingchen Jiang [1] and Jialong Sun [1]

[1] School of Marine Technology and Geomatics, Jiangsu Ocean University, Lianyungang 222005, China; 2021220403@jou.edu.cn (W.C.); 2022220207@jou.edu.cn (J.C.); jiangtc@jou.edu.cn (T.J.); 2006000076@jou.edu.cn (J.S.)

[2] Lianyungang Water Conservancy Planning and Design Institute Co., Ltd., Lianyungang 222006, China; lygslsjy@163.com

* Correspondence: wangxiao@jou.edu.cn; Tel.: +86-137-7548-9830

**Abstract:** A multibeam water column image (WCI) can provide detailed seabed information and is an important means of underwater target detection. However, gas plume targets in an image have no obvious contour information and are susceptible to the influence of underwater environments, equipment noises, and other factors, resulting in varied shapes and sizes. Compared with traditional detection methods, this paper proposes an improved YOLOv7 (You Only Look Once vision 7) network structure for detecting gas plume targets in a WCI. Firstly, Fused-MBConv is used to replace all convolutional blocks in the ELAN (Efficient Layer Aggregation Networks) module to form the ELAN-F (ELAN based on the Fused-MBConv block) module, which accelerates model convergence. Additionally, based on the ELAN-F module, MBConv is used to replace the $3 \times 3$ convolutional blocks to form the ELAN-M (ELAN based on the MBConv block) module, which reduces the number of model parameters. Both ELAN-F and ELAN-M modules are deep residual aggregation structures used to fuse multilevel features and enhance information expression. Furthermore, the ELAN-F1M3 (ELAN based on one Fused-MBConv block and three MBConv blocks) backbone network structure is designed to fully leverage the efficiency of the ELAN-F and ELAN-M modules. Finally, the SimAM attention block is added into the neck network to guide the network to pay more attention to the feature information related to the gas plume target at different scales and to improve model robustness. Experimental results show that this method can accurately detect gas plume targets in a complex WCI and has greatly improved performance compared to the baseline.

**Keywords:** multibeam water column image; gas plume; target detection; YOLOv7; deep residual aggregation structure; SimAM block

## 1. Introduction

Multibeam echo sounding (MBES) is a high-precision underwater data measurement technique [1]. Compared with the traditional single-beam echo sounding technique, MBES uses multiple transmitters and receivers to collect echo signals in multiple directions simultaneously, obtaining more precise water depth and water body data. This technique has been widely applied in marine resource exploration [2], underwater pipeline laying [3], and underwater environmental monitoring [4]. The water column image (WCI) formed by water body data is an important means of underwater target detection, and the gas plumes in the water may be an indication of the presence of gas hydrates in the nearby seabed sediment layers [5]. Development and excavation of these resources will play a crucial role in alleviating current global issues such as energy scarcity and environmental degradation. Therefore, how to detect and locate gas plumes quickly and accurately has become an important research topic.

In traditional WCI target detection, the processing steps are image denoising, feature extraction, and target classification, in that order. Due to the influence of the working principle of MBES, some sidelobe beams are produced around the main lobe beam during transmission. When receiving the echo signal, the echo information of these sidelobe beams is mistaken for real signals, causing significant arc noise in the image [6], which is the main factor affecting image quality. In addition, the image also includes multisector noise and environmental noise. To make the collected raw data more accurate, [7] used weighted least squares to estimate the optimal beam incidence angle and corrected the difference in echo intensity under different water depths, thus obtaining normalized echo data. To effectively remove noise in the image, the image masking method was used to eliminate static noise interference in [8], and artificial thresholding was used to remove environmental noise. In [9], the data that fell within the minimum slant range were considered valid and used in the analysis, and the noise was removed by using the average echo intensity as the threshold. The second step in WCI target detection is to extract features from the denoised image. Feature extraction aims to obtain distinctive and representative image features, such as edges, morphology, and texture, to reduce data dimensions for subsequent classification. In [10], the authors used a clustering algorithm to extract information about regions containing gas plumes and then used feature histograms for feature matching to identify them. In [11], gas plume features were extracted using intensity and morphological characteristics to distinguish them from the surrounding environment. In [12], multiple features, such as color, gradient, and direction, were used for feature extraction, obtaining a set of feature vectors that can effectively distinguish between textures and nontextures. The final step is target classification. Based on the principle of feature invariance, the extracted features are input into a classifier for training. Commonly used classifiers include SVM (Support Vector Machine), Adaboost (Adaptive Boosting), and Random Forest. Then the training results are evaluated and optimized to achieve the high-precision detection of targets in the image. Traditional target detection methods for a WCI are complex, with human factors having a significant impact on image denoising and feature extraction algorithms being unable to extract representative features in complex images. The classifiers used in target classification are strongly influenced by lighting, angle, and noise, making it easy to miss or misidentify targets. Overall, target detection using traditional methods in a WCI is highly limited.

Convolutional neural networks (CNN) have proven effective for solving various visual tasks [13–15] in recent years, thus providing a new solution for multibeam WCI target detection. Compared to various machine learning classifiers, CNN not only automatically extracts image features, reducing human intervention, but can also learn more complex features, improving the model's robustness. In addition, the end-to-end advantage and the introduction of transfer learning [16] have increased the efficiency and accuracy of the model, and it has been applied to different scenarios and tasks. CNN detectors can be divided into one-stage and two-stage, which differ mainly in the order of detection and classification. One-stage detectors refer to the prediction of target category and position directly from feature maps, such as YOLO (You Only Look Once) [17–21], SSD (Single-Shot Multi-Box Detector) [22], RetinaNet [23], and EfficientDet [24]. Among them, YOLO adopts a multiscale feature map and anchor mechanism, which can detect multiple targets simultaneously. SSD adopts feature maps of different scales and multiple detection heads, which can detect targets of different sizes. The focal loss function is used in RetinaNet to reduce the effect of target class imbalance and has achieved excellent detection results. EfficientDet adopts a scalable convolutional neural network structure based on EfficientNet [25] and performs well in speed and accuracy. In [26], to adapt to the particularity of the underwater environment, the author introduced the CBAM (Convolutional Block Attention Module) based on YOLOv4 to help find attention regions in object-dense scenes. In [27], the author established direct connections between different levels of feature pyramid networks to better utilize low-level feature information, thereby increasing the capability and accuracy of feature representation. To make the model smaller, in [28], by pruning and fine-tuning

the EfficientDet-d2 model, the author achieved a 50% reduction in model size without sacrificing detection accuracy. Two-stage detector initially generates candidate frames and then uses them for classification and position regression, such as Faster R-CNN (Region-based Convolutional Neural Network) [29], Mask R-CNN [30], and Cascade R-CNN [31]. Among them, Faster R-CNN uses the RPN (Region Proposal Network) to generate candidate boxes and then uses RoI (Region of Interest) pooling to extract features for classification and regression. Mask R-CNN extends Faster R-CNN by incorporating segmentation tasks, enabling it to perform target detection and instance segmentation simultaneously. Cascade R-CNN enhances the robustness and accuracy of the detector through a cascaded classifier. Wang [32] enhanced the Faster R-CNN algorithm by implementing automatic selection of difficult samples for training, thereby improving the model's ability to perform well and generalize on difficult samples. In [33], Song proposed Boosting R-CNN, a two-stage underwater detector featuring a new region proposal network for generating high-quality candidate boxes.

These methods have shown good improvements in their respective tasks but may not be applicable to gas plume object detection. This is because a gas plume generally consists of numerous bubbles, which are close together and interfere with each other. Compared to other objects, the reflection intensity of the gas plume is weaker, and the edges of the bubbles are not easily distinguishable. In addition, gas plumes in water often experience fracturing as they rise, which makes them difficult to accurately locate. This paper demonstrates through extensive experiments that the proposed method offers a viable solution to address the issues related to gas plume target detection. The main contributions are as follows:

1. ELAN-M and ELAN-F modules are designed to reduce model parameters, speed up model convergence, and alleviate the problem of insignificant accuracy gains in deep residual structures.
2. An ELAN-F1M3 backbone network structure is designed to fully utilize the efficiency of the ELAN-F and ELAN-M modules.
3. To reduce the effect of noise, the SimAM module is introduced to evaluate the weights of the neurons in each feature map of the neck network.
4. Extensive experiments show that the new model can accurately detect plume targets in complex water images, far outperforming other models in terms of accuracy.

## 2. Related Work

### 2.1. Data Augmentation

Data augmentation is a powerful technique used in neural networks to enhance the quantity and diversity of the training data by applying random transformations. By doing so, the model becomes more robust and better equipped to generalize to unseen data. This can alleviate the overfitting problem to some extent. Common augmentation methods include geometric transformations and color transformations. Geometric transformations include flipping, translation, rotation, scaling, etc., which have small changes in the original content. Color transformations include brightness, saturation, color inversion, histogram equalization, etc., which have significant changes in the original content and high diversity. In addition, there are a number of unique approaches being used to augment data. Some scholars have proposed data augmentation methods based on multisample interpolation, such as Sample Pairing [34], Mixup [35], Mosaic [36], etc. Sample Pairing randomly selects two images from the training set, averages the pixel values to synthesize a new sample, and uses one of the image labels as the correct label for the synthesized image; Mixup is an extension of Sample Pairing, which performs linear interpolation on both images and labels; Mosaic combines four training images in a way that is randomly scaled, cropped, and arranged to improve its ability to identify small targets. Some examples of WCI data augmentation are shown in Figure 1.

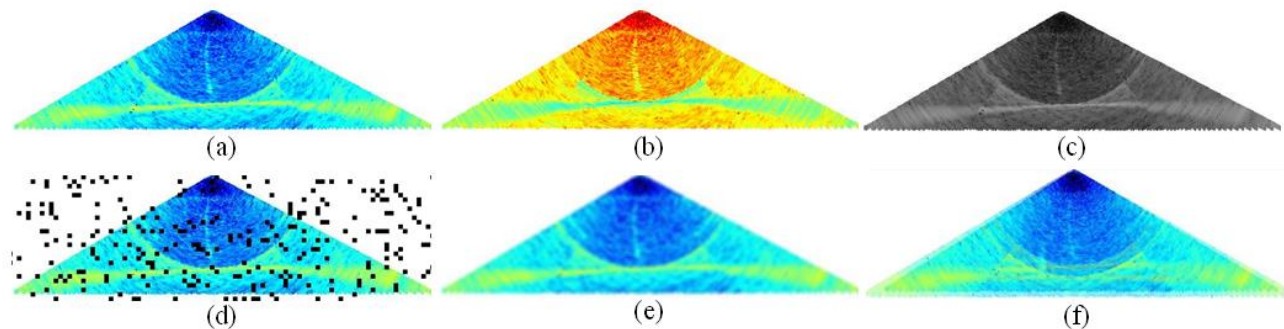

**Figure 1.** Partial results of data augmentation. (**a**) Original image. (**b**) Color transformation. (**c**) Grayscale transformation. (**d**) Random occlusion. (**e**) Blurring. (**f**) Mixup.

### 2.2. YOLOv7

The YOLOv7 model in the one-stage detector is another great achievement in the YOLO family, integrating E-ELAN (Extended Efficient Layer Aggregation Networks), structural reparameterization [37], positive and negative sample allocation strategies [17,18], and a training method with auxiliary heads, once again balancing the contradictions between the number of parameters, computational complexity, and performance. YOLOv7 has seven different versions, including YOLOv7-tiny, YOLOv7, YOLOv7x, YOLOv7-w6, YOLOv7-d6, YOLOv7-e6, and YOLOv7-e6e. Among them, YOLOv7-tiny, YOLOv7, and YOLOv7-w6 are the basic models of the network, and the other models are obtained by model scaling.

As shown in Figure 2, the YOLOv7 network structure consists of an input module, a backbone network, a neck network, and a head network.

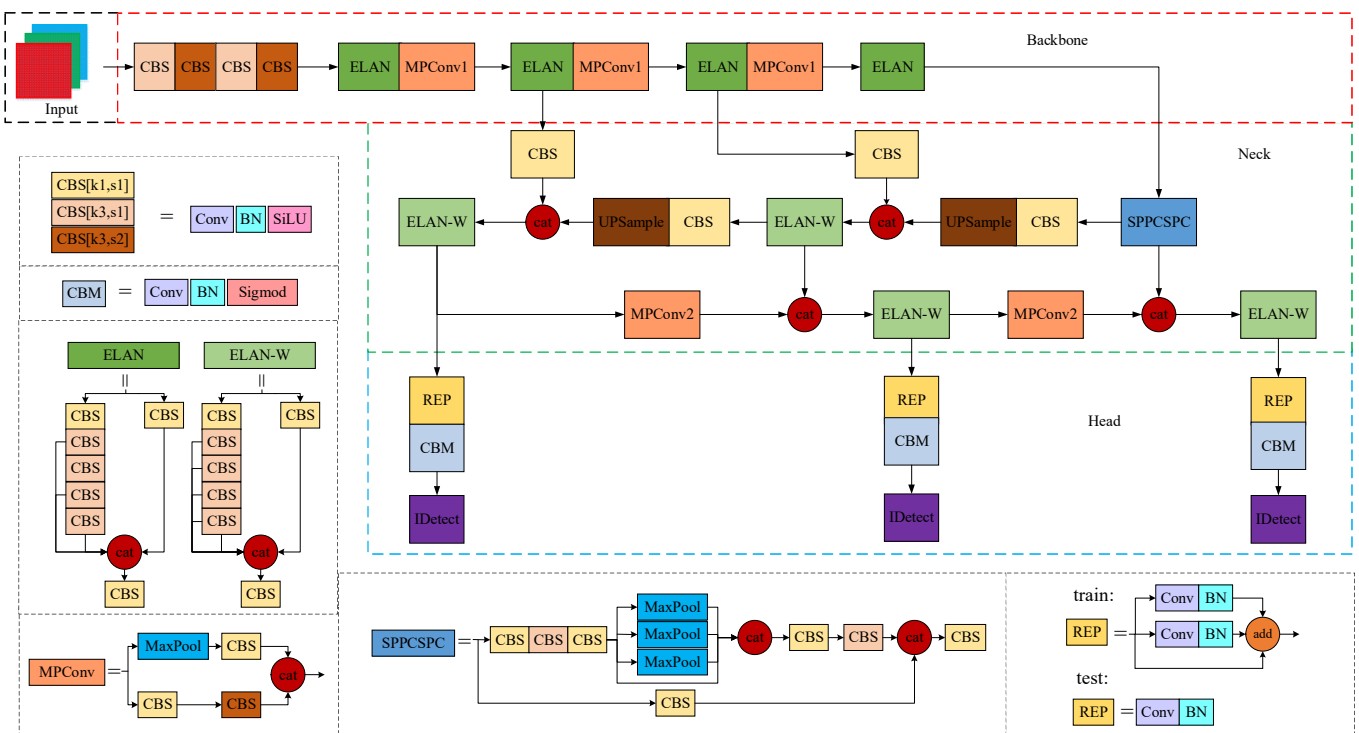

**Figure 2.** Structure of the YOLOv7.

### 2.2.1. Input Module

The input end of YOLOv7 continues to use the improvement points of YOLOv5, mainly utilizing the mosaic high-order data augmentation strategy to increase data diversity and

reduce the computation cost of training. In addition, YOLOv7 uses an adaptive image adjustment strategy to calculate the input size of images. After calculating, the image is adaptively padded on all sides to obtain the final input image, thereby reducing the problem of increased inference time caused by excessive invalid information introduced by conventional image scaling and padding.

### 2.2.2. Backbone Network

The backbone network of YOLOv7 mainly consists of three types of modules: CBS, ELAN [38], and MPConv1. The CBS module includes Convolution (Conv), Batch Normalization (BN), and an SiLU activation function (k and s represent the size and stride of the convolution kernel). The ELAN module is a multibranch structure that effectively reduces the number of neurons in the network through multilayer aggregation, reducing computational and storage overheads, and by controlling the shortest and longest gradient paths to accelerate gradient propagation. Except for the first two $1 \times 1$ convolution kernel sizes of the CBS module, which achieve channel compression, the number of input and output channels of the remaining CBS modules is kept consistent, which has been proven to be an efficient network design principle in Shufflenet v2 [39]. The E-ELAN structure proposed in YOLOv7 is a grouped convolution of two ELAN structures with a group number of two, and the output results are concatenated in the channel direction. The MPConv1 structure is a two-branch structure composed of MaxPool (MP) and CBS, where one branch implements spatial downsampling through MP, and the other branch uses a $3 \times 3$ convolution with a stride of 2 to complete the downsampling. Finally, an enhanced version of the downsampling function is implemented through the connection operation. Both ELAN and MPConv1 are a sublimation of feature reuse, making the network better at capturing relationships between data.

### 2.2.3. Neck Network

The neck network of YOLOv7 introduces the SPPCSPC (Spatial Pyramid Pooling and Cross-Stage Partial Connection) module, which expands the receptive field and achieves multiscale feature fusion through MP operations with different pooling kernels. Immediately afterward, the enhanced feature extraction network structure of Feature Pyramid Network (FPN) + Pyramid Attention Network (PAN) is still adopted. The FPN + PAN architecture improves target discrimination at different scales by combining bottom-up feature extraction with top-down feature fusion. The ELAN-W and MPConv2 modules used here are similar to the ELAN and MPConv1 modules used in the backbone network. ELAN-W is simply an extension of ELAN, with the addition of two outputs in one of its branching structures for later concatenation; MPConv2 uses the same input and output channel numbers in the CBS module.

### 2.2.4. Head Network

The prediction part of YOLOv7 uses the reparameterization technique of RepVGG [37]. During model training, the REP structure consists of a $3 \times 3$ convolution branch for feature extraction, a $1 \times 1$ convolution branch for smoothing features, and an identity transformation branch. The three branches are combined through connection operation, improving network performance. In the inference stage, the REP structure is reparametrized into a $3 \times 3$ convolution operation to reduce model parameters and accelerate inference speed. In the prediction stage, the auxiliary head is used to supervise the output of the intermediate layer features so that the intermediate layer can more accurately represent the information in the input data, thus improving the expressiveness of the model.

## 3. Method

### 3.1. MBConv and Fused-MBConv Block

To achieve higher detection accuracy, we typically use deeper network models to capture more complex feature information. However, this often leads to longer train-

ing times, overfitting, gradient vanishing, and gradient exploding problems. Although residual connections [40] effectively solve these problems, the entire network model still requires significant resources. Therefore, a lightweight model has always been the goal of many researchers.

MobileNetV2 [41] introduces two modules: the linear bottleneck and the inverted residual. The linear bottleneck is a bottleneck block that removes the last activation function to avoid the loss of feature information; the inverted residual is used to form sparse features and reduce loss by first up-dimensioning and then down-dimensioning operations and then reducing model parameters and extracting high-dimensional features using depthwise convolution (DWConv). The combination of the two is called MBConv in other applications [42–44], and in later applications, channel attention [45] was also added. In Figure 3, the MBConv is composed of an expand convolution with BN and SiLU activation function, a DWConv with BN and SiLU activation function for parameter reduction, a channel attention block for calculating channel weights, a low-rank project convolution with BN, and a dropout layer. The channel attention block (Figure 4) consists of two operations: squeeze and excitation (SE). The squeeze operation uses Global Average Pooling (GAP) to compress the size of the feature map from $H \times W \times C$ to $1 \times 1 \times C$. This process compresses the height and width of each channel into a real number with global information, allowing the overall model to significantly reduce its number of parameters while preserving global features. The excitation operation first applies a fully connected (FC) layer to compress the channels (r denotes the compression ratio) to reduce the number of channels to further reduce the computational complexity; then after activation using the Relu activation function, the number of channels is restored to the original dimension by a second FC, followed by the Sigmoid activation function to obtain the final weight (different colors represent different weight values) to distinguish the importance of different channels. Finally, the total number of output features is obtained by multiplying the output weight coefficients on the branch with the original feature values of the model.

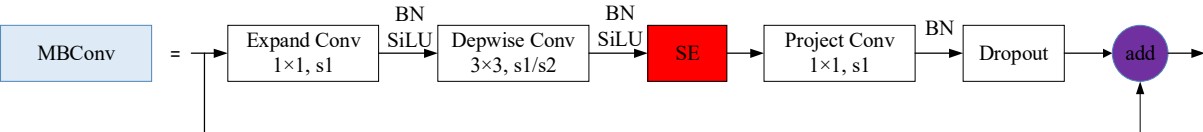

**Figure 3.** Structure of the MBConv.

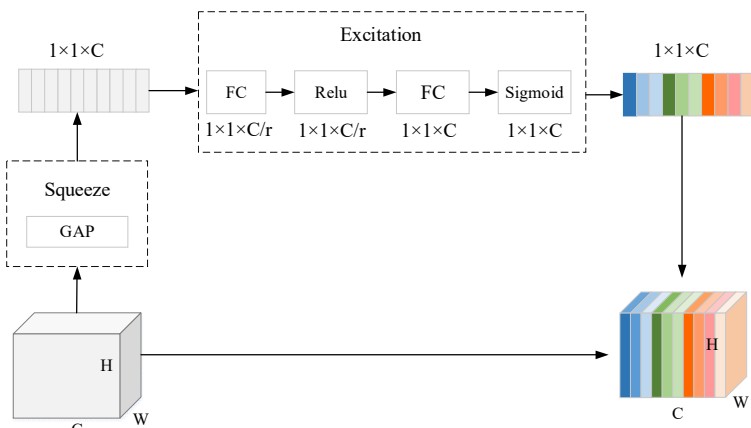

**Figure 4.** Structure of the squeeze and excitation operation.

Based on MBConv, Fused-MBConv is proposed. The authors of EfficientNet v2 [44] found that although DWConv can theoretically reduce the number of model training parameters, in practice it is slow to use on shallow networks, does not achieve the desired state, and does not fully utilize existing accelerators. Therefore, in the structure of MBConv,

the DWConv block is removed. When channel expansion is not performed, a $3 \times 3$ ordinary convolution is used to replace the original expand convolution block, the SE block, and the low-rank project convolution block. When channel expansion is performed, only the original SE block is removed. The structural diagram of Fused-MBConv is shown in Figure 5.

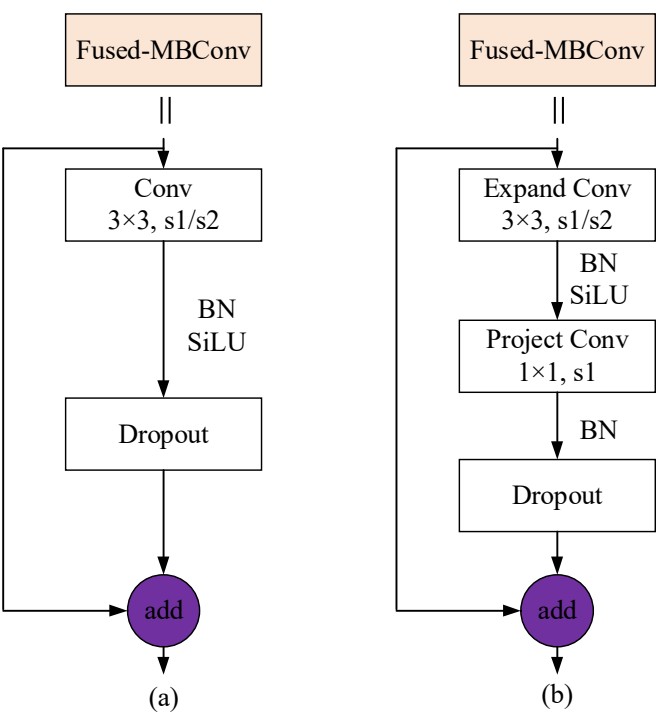

**Figure 5.** Structure of the Fused-MBConv: (**a**) the structure without channel expansion; (**b**) the structure with channel expansion.

### 3.2. ELAN-F and ELAN-M Module

To maximize the advantages of the MBConv block, Fused-MBConv block, and YOLOv7 itself, this paper embeds the MBConv and Fused-MBConv blocks into the ELAN structure of the backbone network and constructs the ELAN-F (ELAN based on the Fused-MBConv block) acceleration convergence module and the ELAN-M (ELAN based on the MBConv block) parameter reduction module (In Figure 6, c represents the number of channels). In the ELAN-F module, the original CBS block is replaced by the Fused-MBConv block, while the ELAN-M module is based on the ELAN-F, where the convolution kernel size of $3 \times 3$ is replaced by the MBConv block. In EfficientNet v2, the authors set the convolutional kernel sizes of both Fused-MBConv and MBConv to $3 \times 3$, while in the ELAN structure of this paper, the remaining $1 \times 1$ convolutional kernels of the CBS block are replaced by the Fused-MBConv block without channel expansion. The aim is to speed up model convergence and improve network accuracy without increasing the number of network parameters. Both the ELAN-F module and ELAN-M module are multibranch deep residual aggregation structures. After replacing the Fused-MBConv and MBConv blocks, the network depth is greatly increased, and deep semantic information is extracted through multiple residual connections. However, in general, as the depth of the network increases, the residual become less and less effective. In this paper, these two residual structures are added to the multilayer aggregation structure of ELAN, and the neurons of different layers are connected again through cross-layer connections to achieve information sharing, which alleviates the problem of deep residuals and enables the network to perform with better accuracy thanks to the combination of residual connections and multilayer aggregation.

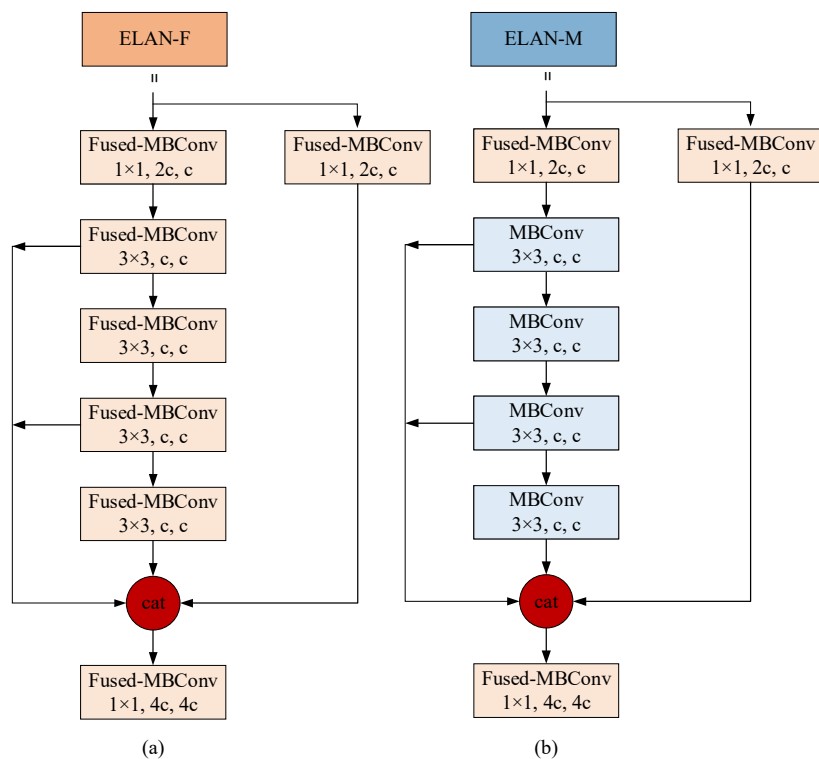

**Figure 6.** Structure of the ELAN-F and ELAN-M: (**a**) ELAN-F, (**b**) ELAN-M.

### 3.3. SimAM Attention Block

Many researchers [46–48] have demonstrated the effectiveness of attention mechanisms in helping models understand important features in images, reducing noise interference, and improving model robustness. As a plug-and-play block, it can be quickly applied at different positions in different networks, demonstrating its simplicity. SimAM is an attention block proposed by Yang [49] that has three-dimensional (3D) weights, as shown in Figure 7. By simultaneously considering the relationship between space and channels, the 3D weights of the neurons are generated and are assigned to the original feature map.

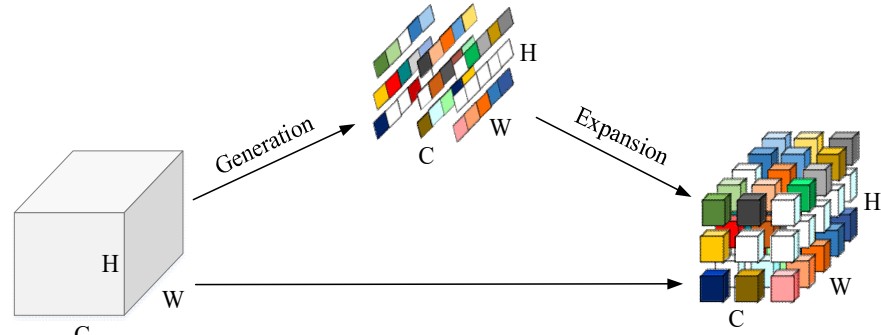

**Figure 7.** Structure of the SimAM attention block with 3D weights.

In neuroscience, neurons with important information often exhibit a different firing pattern than surrounding neurons and suppress the activity of surrounding neurons. Based on this, Yang defines an energy function for each neuron in the feature map to distinguish the target neuron from other neurons. The function (1) is shown below:

$$e_t(w_t, b_t, y, x_i) = (y_t - \hat{t})^2 + \frac{1}{M-1} \sum_{i=1}^{M-1} (y_0 - \hat{x}_i)^2 \tag{1}$$

where t and $x_i$ are the target neurons and other neurons in a single channel in the input feature map X. $\hat{t}$ and $\hat{x}_i$ are obtained by a linear transformation of t and $x_i$ with the transformation equations $\hat{t} = w_i t + b_t$ and $x_i = w_t x_i + b_t$, where i is the index on the spatial dimension and $M = H \times W$ is the number of neurons on that channel, and $w_t$ and $b_t$ are the weights and biases of the transformations. By solving for the minimum of (1), the linear separability of the target neuron t from all other neurons in the same channel can be obtained. For $y_t$ and $y_0$, using binary labels and adding the regularizer $\lambda w_t^2$ to (1), the transformed energy expression is

$$
\begin{aligned}
e_t(w_t, b_t, y, x_i) = & \frac{1}{M-1} \sum_{i=1}^{M-1} \left( -1 - (w_t x_i + b_t) \right)^2 \\
& + (1 - (w_t t + b_t))^2 + \lambda w_t^2
\end{aligned}
\tag{2}
$$

Equation (2) is a closed-form solution concerning $w_t$ and $b_t$. The analytic equations for $w_t$ and $b_t$ are

$$
w_t = -\frac{2(t - u_t)}{(t - u_t)^2 + 2\sigma_t^2 + 2\lambda}
\tag{3}
$$

$$
b_t = -\frac{1}{2}(t + \mu_t) w_t
\tag{4}
$$

Assuming that all pixels in each channel follow the same distribution and that the mean $\mu_t = (1/(M-1)) \sum_1^{M-1} x_i$ and variance $\sigma_t^2 = (1/(M-1)) \sum_i^{M-1} (x_i - \mu_t)^2$ are known for all neurons except t, the minimizing neuron energy function is

$$
e_t^* = \frac{4(\hat{\sigma}^2 + \lambda)}{(t - \hat{\mu})^2 + 2\hat{\sigma}^2 + 2\lambda}
\tag{5}
$$

where $\hat{\mu} = (1/M) \sum_{i=1}^{M} x_i$ and $\hat{\sigma}^2 = (1/M) \sum_{i=1}^{M} (x_i - \hat{\mu})^2$. When $e^*$ is smaller, it means that neuron t is more distinct from peripheral neurons and that neuron t is more important and should be given a higher weight. Thus, the importance of each neuron can be obtained by $1/e_t^*$. Finally, feature refinement is carried out through the sigmoid function, and the entire refinement phase is

$$
\widetilde{X} = \text{sigmoid}\left(\frac{1}{E}\right) \odot X
\tag{6}
$$

where E denotes the grouping of all $e_t^*$ in the spatial and channel dimensions, and $\odot$ indicates the multiplication operation.

The SimAM attention block obtains the 3D weights of each neuron by optimizing the energy function, avoiding the structural tuning work of other similar attention blocks. In addition, its parameter-free nature results in a minimal computational overhead.

### 3.4. YOLOv7 Neck Network with SimAM

In Figure 8, The CBS modules are replaced by SimAM in two downsampling blocks of the neck network. The SimAM module reduces feature loss during downsampling by re-evaluating each neuron in the feature map. Moreover, MPConv2 is located in the PAN structure of the neck network, which includes feature maps of multiple scales. Different scales of feature maps contain information about objects of different scales. The SimAM attention block can enhance the interaction and weight adjustment between different scales of feature maps, thus better capturing the features of target areas.

After replacing the ELAN-F, ELAN-M, and SimAM modules in YOLOv7, the proposed YOLOv7-FMS network structure is obtained. In the target detection task for gas plumes, which have scarce information and unclear contours, the YOLOv7-FMS network can fully use the extracted features and reduce feature loss during the feature extraction process, enabling the network to quickly and accurately locate useful regions in complex images and improve detection accuracy.

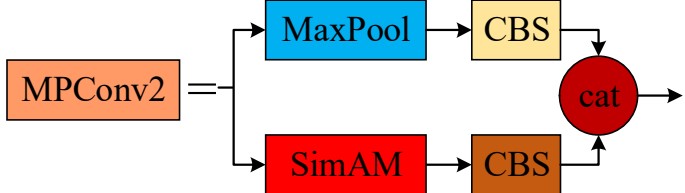

**Figure 8.** Structure of the SimAM module in MPConv2.

## 4. Experiments and Discussion

### 4.1. Preliminary Preparation

#### 4.1.1. Dataset Preparation

This paper uses the Kongsberg EM710 multibeam bathymetry system to make in situ measurements in a marine area, obtaining a total of 320 images containing gas plume targets. During the data augmentation process, relevant studies [50,51] have shown that if the overall dataset is augmented first, the augmented data of the same image may be split into the training and validation sets during data splitting. This could cause the model to become overconfident and degrade its generalization ability. Therefore, this paper first divided the dataset in an 8:1:1 ratio, and then data augmentation was performed. The augmented dataset consisted of 1920 images, with 1536, 192, and 192 images used for training, validation, and testing. The dataset is labeled in YOLO format using LabelImg software.

#### 4.1.2. Experimental Environment

This study was conducted on an Ubuntu 20.04 operating system with an Intel Xeon Platinum 8255C processor and an RTX 3090 (24 GB) graphics card. The GPU acceleration environment was created using CUDA 11.3, and the network framework was built using Python 3.8 and PyTorch 1.11.0. The development platform was Visual Studio Code 1.75.1.

#### 4.1.3. Hyperparameter Setting

All experiments were performed with the same hyperparameters, which are listed in Table 1, to demonstrate the effectiveness of our method.

**Table 1.** Hyperparameter configuration.

| Hyperparameter | Configuration |
| --- | --- |
| Initial learn rate | 0.01 |
| Optimizer | SGD |
| Weight decay | 0.0005 |
| Momentum | 0.937 |
| Image size | $320 \times 320$ |
| Batch size | 16 |
| Epochs | 400 |

### 4.2. Model Evaluation Metrics

This study used parameters, computational complexity, FPS (Frames Per Second), F1, and mAP (mean Average Precision) to evaluate the performance of each model. Parameters and computational complexity measure the spatial and temporal complexity of a model, representing the number of learnable parameters and floating point operations, respectively; FPS refers to how many images the model can detect per second. F1 is a single score that evaluates the model and is a weighted average that considers precision and recall. Precision and recall indicate the proportion of false detections and missed detections in the dataset, respectively. Average precision (AP) indicates the performance of the model in a single category. mAP is the mean of AP for all categories, and we only studied the detection of gas plume, so AP is equal to mAP. mAP50 represents the mAP calculated using an

IoU (Intersection over Union) threshold of 0.50; mAP50:95 represents a set of mAP values calculated using multiple IoU thresholds from 0.50 to 0.95 and then averaged to full evaluation of model performance. The formulae are as follows, where TP (True Positive), FP (False Positive), and FN (False Negative) indicate the number of gas plume targets detected correctly, incorrectly, and not detected in the water column image.

$$\text{Precision} = \frac{\text{TP}}{\text{TP} + \text{FP}} \tag{7}$$

$$\text{Recall} = \frac{\text{TP}}{\text{TP} + \text{FN}} \tag{8}$$

$$\text{F1} = \frac{2 \times \text{Precision} \times \text{Recall}}{\text{Precision} + \text{Recall}} \tag{9}$$

$$\text{AP} = \int_0^1 \text{Precision}(\text{Recall})\text{dRecall} \tag{10}$$

### 4.3. Experimental Analysis

To verify the effectiveness of the YOLOv7-FMS model, a series of comparison and ablation experiments was conducted, and the results were compared in detail using the previously defined evaluation metrics.

#### 4.3.1. The Selection of the Baseline Network

In the YOLOv7 model architecture, the three network structures selected for comparison were YOLOv7-tiny, YOLOv7, and YOLOv7x. From Table 2, it can be seen that although YOLOv7-tiny has a smaller number of parameters, a higher number of FLOPs, and a higher FPS, its accuracy is minimal and cannot meet the testing requirements. Although YOLOv7x has similar accuracy to YOLOv7, it consumes more resources and has the slowest inference speed. After comparing these models, YOLOv7 was chosen as our baseline network for further improvements.

**Table 2.** Performance Comparison of Different Models of YOLOv7 Network.

| Method | Params. (M) | FLOPs (G) | F1 (%) | mAP50 (%) | mAP50:95 (%) | FPS Batch_Size = 1 |
|---|---|---|---|---|---|---|
| YOLOv7-tiny | 6.0 | 13.0 | 82.9 | 83.0 | 38.2 | 128.2 |
| YOLOv7x | 70.8 | 188.9 | 89.1 | 90.8 | 44.1 | 78.4 |
| YOLOv7 | 36.5 | 103.2 | 90.8 | 91.0 | 46.8 | 84.7 |

#### 4.3.2. Design of Backbone Network Based on ELAN-F and ELAN-M

The backbone of YOLOv7 contains four ELAN modules, which are replaced by the ELAN-F and ELAN-M modules. The new backbone network is designed in Figure 9, and the performance is shown in Table 3. The YOLOv7 backbone network with ELAN-F1M3 has the highest accuracy, with F1, mAP50, and mAP50:95 increasing by 3.6, 6.7, and 8.4%, respectively, compared to the baseline. Furthermore, the numbers of parameters and FLOPs are reduced by 16.7 and 14.2%. However, the inference speed of 57.8 is the slowest because the ELAN-M module uses DWConv, which wastes some time on reading and writing data from memory, and the GPU's computing power is not fully utilized. Moreover, even the worst YOLOv7-F2M2 has nearly the same accuracy as the YOLOv7 baseline network. To verify the accelerated convergence function of the model, Figure 10 shows the mAP50 change curve of these network structures throughout the training process. The curve results show that YOLOv7-F1M3 can achieve high accuracy in fewer batches and gradually become stable. This indicates that using the ELAN-F module only in the shallow layers of the network can achieve the best results, which is consistent with the conclusion of Efficient

v2 obtained through NAS (Neural Architecture Search), where the Fused-MBConv module is used only in the shallow layers of the network.

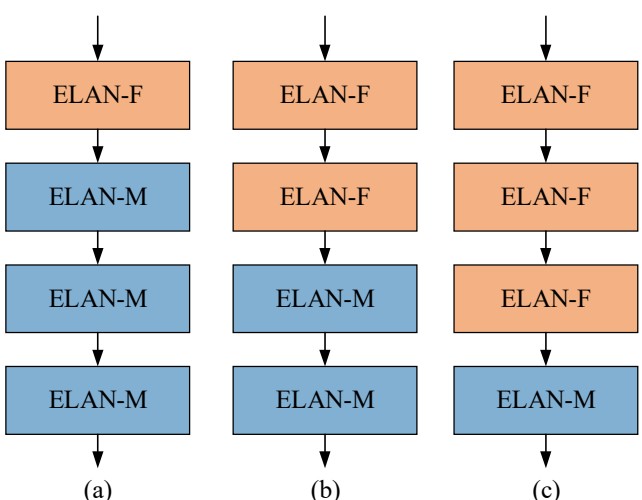

**Figure 9.** Comparison of Backbone Networks with Different Combinations. (**a**) One Fused-MBConv block and Three MBConv blocks (i.e., ELAN-F1M3). (**b**) Two Fused-MBConv blocks and Two MBConv blocks (i.e., ELAN-F2M2). (**c**) Three Fused-MBConv blocks and One MBConv block (i.e., ELAN-F3M1).

**Table 3.** Performance Comparison of Backbone Networks with Different Configurations.

| Method | Params. (M) | FLOPs (G) | F1 (%) | mAP50 (%) | mAP50:95 (%) | FPS Batch_Size = 1 |
|---|---|---|---|---|---|---|
| YOLOv7-F1M3 | 30.4 | 88.5 | 94.4 | 97.7 | 55.2 | 55.6 |
| YOLOv7-F2M2 | 30.9 | 95.1 | 89.5 | 91.5 | 46.2 | 62.9 |
| YOLOv7-F3M1 | 33.1 | 101.6 | 90.7 | 93.6 | 52.2 | 68.5 |

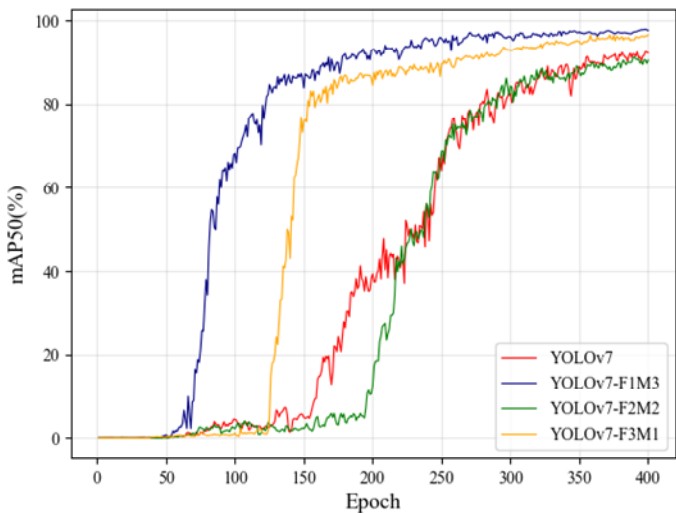

**Figure 10.** Comparison of the mAP curves of backbone networks under different configurations.

### 4.3.3. Experimental Analysis of the Proposed Method and Other Advanced Lightweight Networks

Table 4 shows the comparative results of the YOLOv7-F1M3 model and other lightweight networks such as MobileNet v3 [52], ShuffleNet v2 [39], GhostNet v2 [53], PP-LCNet (Positional Pyramid-based Lightweight Convolutional Network) [54], and MobileOne [55] by

replacing the backbone network of the baseline. Although lightweight networks reduce the numbers of parameters and FLOPs, the corresponding feature representation capacity is reduced, resulting in a significant decrease in accuracy. Among them, MobileOne achieves the same lowest number of parameters and highest detection speed by implementing a branch-free network structure through reparameterization in the inference process. Although the YOLOv7-F1M3 has relatively high number of parameters and FLOPs, it has the highest detection accuracy, outperforming the second-placed GhostNet v2 by 3.1, 6.6, and 10.2% on F1, mAP50, and mAP50:95, respectively.

**Table 4.** Performance Comparison of Different Advanced Lightweight Networks in the Backbone Network.

| Method | Params. (M) | FLOPs (G) | F1 (%) | mAP50 (%) | mAP50:95 (%) | FPS Batch_Size = 1 |
|---|---|---|---|---|---|---|
| YOLOv7-MobileNetv3 | 24.8 | 36.9 | 80.1 | 82.5 | 35.7 | 22.6 |
| YOLOv7-ShuffleNetv2 | 23.3 | 37.9 | 83.0 | 84.6 | 35.7 | 74.1 |
| YOLOv7-GhostNetv2 | 29.6 | 75.2 | 91.3 | 91.1 | 45.0 | 38.5 |
| YOLOv7-PPLCNet | 28.9 | 63.9 | 83.9 | 86.1 | 43.5 | 63.3 |
| YOLOv7-MobileOne | 23.3 | 40.1 | 87.6 | 89.4 | 39.5 | 90.9 |
| YOLOv7-F1M3 | 30.4 | 88.5 | 94.4 | 97.7 | 55.2 | 55.6 |

### 4.3.4. Experimental Analysis of the Proposed Method and Other Attention Blocks

In Table 5, the performance of SimAM in the baseline neck network is compared with other attention mechanisms, including SE [45], ECA (Efficient Channel Attention) [46], CBAM [47], and CA (Coordinate Attention) [48]. Compared to the baseline, the detection accuracy improves after the integration of ECA and SimAM. ECA, an improved version of SE, computes channel weights through a learnable 1D convolution kernel, avoiding the use of GPA, which does not capture long-range dependencies in the feature map well. The same GPA is used in CBAM, so it exhibits a relatively low detection accuracy. CA calculates attention weights based on the coordinate information of the target, whereas the gas plume target is randomly distributed in the sea and is often fractured and drifting in the water, making it difficult to detect accurately during the test. However, SimAM achieves the greatest improvement by directly considering the 3D weight relationship of the neurons through the energy function, with F1, mAP50, and mAP50:95 increasing by 2.0, 5.3, and 8.5%, respectively, compared to the baseline, with the same lowest number of FLOPs at an intermediate detection speed, In addition, this module does not require any parameters.

**Table 5.** Performance Comparison of Different Attention Mechanisms in the Neck Network.

| Method | Block Params. | FLOPs (G) | F1 (%) | mAP50 (%) | mAP50:95 (%) | FPS Batch_Size = 1 |
|---|---|---|---|---|---|---|
| YOLOv7-SE | 163,840 | 102.9 | 77.0 | 78.7 | 33.2 | 79.4 |
| YOLOv7-ECA | 2 | 102.7 | 91.1 | 94.5 | 52.4 | 56.5 |
| YOLOv7-CBAM | 772 | 102.7 | 86.8 | 89.5 | 45.6 | 45.9 |
| YOLOv7-CA | 247,680 | 103.4 | 83.3 | 85.8 | 38.6 | 41.8 |
| YOLOv7-SimAM | 0 | 102.7 | 92.8 | 96.3 | 55.3 | 53.5 |

### 4.3.5. Ablation Study Based on YOLOv7

In previous sections, a series of horizontal comparisons among various improvements in the YOLOv7-FMS network have been conducted to demonstrate its superiority over other methods. In Table 6, a longitudinal comparison of the ablation experiments is made, where the model shows some improvement over the baseline in all metrics after adding the ELAN-F1M3 module, the SimAM module, or both. Compared to the SimAM module, the ELAN-F1M3 module, a multibranch deep residual aggregation structure, can learn more useful information and has a greater impact on model enhancement. Moreover, the

YOLOv7-FMS reduces its number of parameters and FLOPs by 17.0 and 14.5% and increases F1, mAP50, and mAP50:95 by 4.4, 7.4, and 10.7%, respectively. However, due to the increase in network depth, the side-connection span between the FPN and the three valid feature maps increases, and the dependency between the data is stronger, leading to a decrease in detection speed again. We also compared the performance of the improved module at different network locations. YOLOv7-F1M3 (Neck) refers to the replacement of four ELAN-W modules in the neck network with ELAN-M modules based on YOLOv7-F1M3. The test results show that the detection accuracy decreases significantly with the large reduction in parameters and computational complexity. In addition, the excessive use of DWConv modules slows down the detection speed of the network. YOLOv7-SimAM (Backbone) refers to the insertion of the SimAM attention module at the connection between the backbone network and the neck network. The detection results are slightly lower than YOLOv7, indicating that the SimAM module in the neck network can better capture the relationship between data and improve network performance.

**Table 6.** Ablation Study Based on YOLOv7.

| Method | Params. (M) | FLOPs (G) | F1 (%) | mAP50 (%) | mAP50:95 (%) | FPS Batch_Size = 1 |
|---|---|---|---|---|---|---|
| YOLOv7 | 36.5 | 103.2 | 90.8 | 91.0 | 46.8 | 84.7 |
| YOLOv7-F1M3 (Neck) | 27.0 | 81.8 | 63.4 | 62.3 | 21.2 | 37.5 |
| YOLOv7-F1M3 | 30.4 | 88.5 | 94.4 | 97.7 | 55.2 | 55.6 |
| YOLOv7-SimAM (Backbone) | 35.4 | 102.3 | 87.7 | 89.2 | 44.5 | 51.0 |
| YOLOv7-SimAM | 36.3 | 102.7 | 92.8 | 96.3 | 55.3 | 53.5 |
| YOLOv7-FMS | 30.3 | 88.2 | 95.2 | 98.4 | 57.5 | 44.6 |

In Figure 11, the CAM (Class Activation Mapping) feature visualization technique is used to generate a weighted heat map for the various attention mechanisms to help us better understand and compare the detection performance and decision-making processes of different attention networks. In Figure 11b, the baseline network of YOLOv7 focuses more on the sidelobe noise and the seabed region and too little on the gas plume target. After adding attention to the network, the weights assigned to the gas plume target in the feature map are enhanced, and the coverage and attention of the region are significantly improved after adding SimAM (Figure 11g) compared with the other four mainstream attention mechanisms. Additionally, it effectively suppresses the sidelobe noise and irrelevant features in the image. This suggests that the model has better robustness with the addition of SimAM.

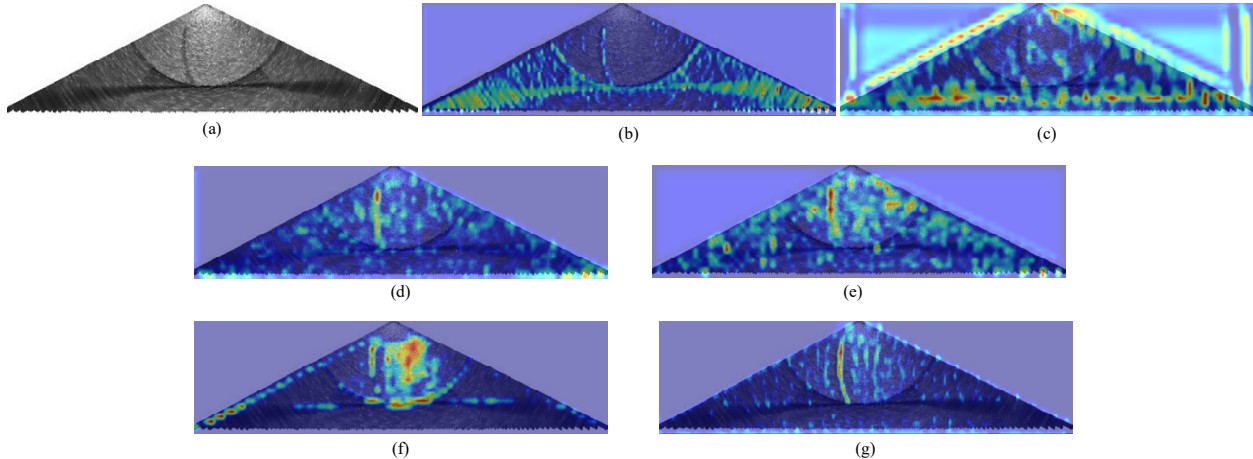

**Figure 11.** Comparison of heat map under different attention mechanisms. (**a**) Original image. (**b**) YOLOv7. (**c**) YOLOv7-SE. (**d**) YOLOv7-ECA. (**e**) YOLOv7-CBAM. (**f**) YOLOv7-CA. (**g**) YOLOv7-SimAM.

### 4.3.6. Experimental Analysis of the Proposed Method and Other CNN Methods

To further validate the performance of the proposed YOLOv7-FMS model, we selected YOLOv5 [17], YOLOX [18], YOLOv6 [19], SSD [22], RetinaNet [23], and EfficientDet [24], all of which are similarly sized detection models, for comparison experiments. Table 7 shows that as the number of model parameters increases, the accuracy of the model also increases. Although the lightweight YOLOv7-FMS model is still at a relatively high level in terms of parameters and computational complexity, in the accuracy metrics of F1, mAP50, and mAP50:95, our approach outperforms the second-ranked YOLOv6-m by 0.8, 0.8, and 7.5%, respectively, indicating that this method has good detection performance even at high detection confidence. The detection accuracy of SSD, RetinaNet, and EfficientDet is relatively low. This is because SSD requires separate prediction of the object's location and category at multiple scales during detection, which may result in some gas plume targets being missed or misclassified, whereas YOLOv7 uses the FPN + PAN structure, as in previous generations, to enable information sharing across multiple scales and pathways. RetinaNet uses focal loss to reduce the weight of easily classified samples in multiclass detection, but it cannot address the problem of size imbalance within the same class of objects. The D4 version of EfficientDet is similar in scale to other models, but its default image input size is 1024, resulting in larger feature map sizes generated during training. This requires more convolution and pooling operations during forward and backward propagation, resulting in increased FLOPs and inference time. In addition, we also trained Faster RCNN [29] and DETR (DEtection TRansformer) [56] on our dataset, but their accuracy was extremely low. The main reason for this is that Faster R-CNN uses too many anchors to generate candidate boxes with RPN, which can easily lead to redundancy. Moreover, for gas plume targets in multibeam water column images, their shapes and features differ greatly from those of other typical targets, making it difficult for RPN to generate sufficiently accurate candidate boxes, which may not be able to adapt to small targets in the gas plumes. Then, due to the relatively small size of the self-built dataset used in this paper and the limited computing resources available, it is difficult to fully leverage the performance of DETR, resulting in difficulty in optimizing the training results.

**Table 7.** Performance Comparison of the Proposed Method and other SOTA Models.

| Method | Params. (M) | FLOPs (G) | F1 (%) | mAP50 (%) | mAP50:95 (%) | FPS Batch_Size = 1 |
|---|---|---|---|---|---|---|
| YOLOv5-m | 21.1 | 12.7 | 92.0 | 96.4 | 50.2 | 70.4 |
| YOLOX-m | 25.3 | 18.4 | 92.3 | 95.1 | 48.2 | 48.7 |
| YOLOv6-m | 34.8 | 21.4 | 94.4 | 97.6 | 50.0 | 67.4 |
| SSD300 | 23.7 | 68.4 | 74.7 | 78.2 | 29.2 | 149.8 |
| RetinaNet (resnet34) | 29.9 | 38.3 | 19.8 | 22.5 | 6.1 | 53.6 |
| EfficientDet-D4 | 20.5 | 104.9 | 69.3 | 71.5 | 23.1 | 12.6 |
| Ours | 30.3 | 88.2 | 95.2 | 98.4 | 57.5 | 44.6 |

### 4.3.7. Result of Detection and Recognition of Gas Plume Targets in WCI

Finally, we test the proposed YOLOv7-FMS model against the original YOLOv7 model on some representative images. Figure 12 shows a clear WCI of the target, where the YOLOv7 FMS bounding box can fit the gas plume target more closely and with better detection accuracy. Figure 13 shows an unclear WCI of the target, where YOLOv7 FMS still detects the target well. In Figure 14, there is an overlapping of targets. The two bounding boxes of YOLOv7-FMS can thus be closer and better represent the morphological of the gas plume target. In Figure 15, the WCI is greatly affected by sidelobe noise, resulting in YOLOv7 missing half of the gas plume targets. Figure 16 shows the phenomenon of gas plume targets fracturing during their upward motion due to the presence of internal waves on the seabed. After detection using YOLOv7, one gas plume target is missed, and two of the targets are mistakenly detected as a single target, whereas YOLOv7-FMS correctly detects all four gas plume targets. The detection results demonstrate that the improved

method can accurately identify gas plume targets in WCI with high noise levels, blurred contours, and complex seabed environments, thereby enhancing the application of the WCI.

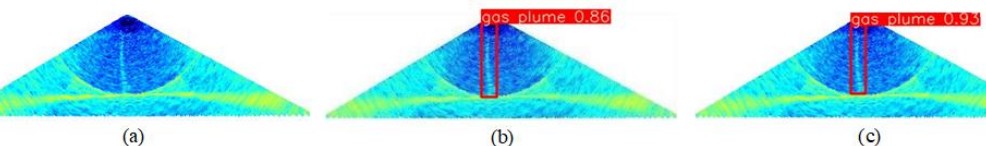

**Figure 12.** Detection and recognition results of the WCI with clear gas plume target: (**a**) original image, (**b**) YOLOv7, (**c**) YOLOv7-FMS.

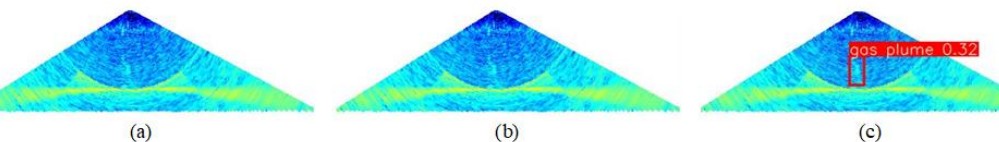

**Figure 13.** Detection and recognition results of the WCI with unclear gas plume target: (**a**) original image, (**b**) YOLOv7, (**c**) YOLOv7-FMS.

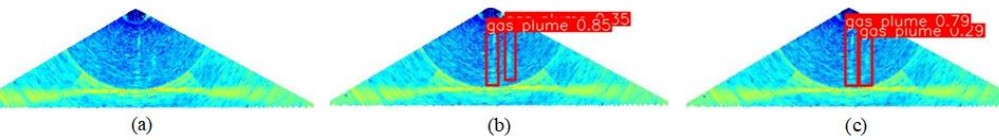

**Figure 14.** Detection and recognition results of the WCI with overlapping gas plume targets: (**a**) original image, (**b**) YOLOv7, (**c**) YOLOv7-FMS.

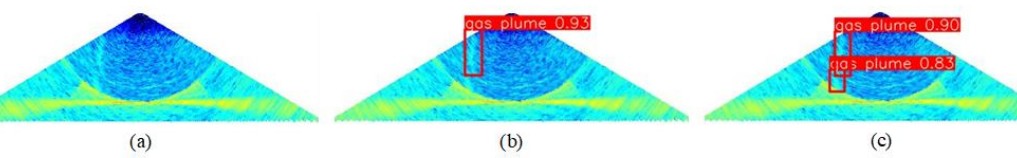

**Figure 15.** Detection and recognition results of the WCI affected by high noise levels: (**a**) original image, (**b**) YOLOv7, (**c**) YOLOv7-FMS.

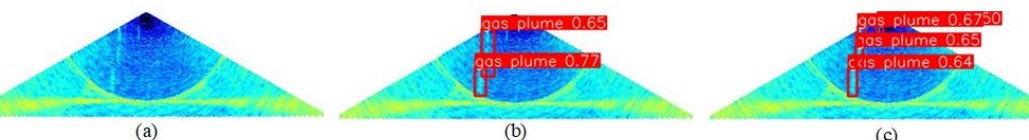

**Figure 16.** Detection and recognition results of the WCI with fractured gas plume targets: (**a**) original image, (**b**) YOLOv7, (**c**) YOLOv7-FMS.

## 5. Conclusions

In this paper, a YOLOv7-FMS model based on the YOLOv7 network structure was proposed. First, in the backbone network, we replaced the ELAN module with the ELAN-F and ELAN-M module and generated the ELAN-F1M3 backbone network, which reduces the numbers of parameters and FLOPs and accelerates the model convergence. The ELAN-F and ELAN-M modules are both internal residual and external aggregation structures. By repeatedly forming cross-layer connections, the model learns more complex mapping functions and reduces information loss. Then, by aggregating the outputs of the internal layers through a multilevel aggregation approach, the feature representation is enriched, and the expressiveness of the model is improved. In addition, the ELAN-M module uses the SE block to enable interaction between channels and enhance feature extraction. Next, we added the SimAM module to the neck network to evaluate the importance of each neuron

in the multiscale feature maps, guided the network to focus on key features, and improved the robustness of the model. Experimental results show that the method outperforms other improvement points of the same type, it can adapt well to the morphological characteristics of the gas plume target, accurately locating the target's position, and that it has a strong anti-interference ability during the detection process. However, due to the significant increase in depth and complexity of the model in the improved network, the detection speed during the detection process has decreased. In future work, we will optimize the network structure through methods such as model pruning and distillation to improve detection speed and efficiency and achieve model deployment.

**Author Contributions:** Conceptualization, W.C. and X.W.; Data curation, W.C. and X.W.; Formal analysis, W.C. and X.W.; Funding acquisition, X.W., B.Y., T.J. and J.S.; Investigation, B.Y. and T.J.; Methodology, W.C. and B.Y.; Project administration, B.Y. and J.C.; Resources, X.W.; Software, W.C. and J.C.; Supervision, T.J. and J.S.; Validation, J.C., T.J. and J.S.; Visualization, W.C. and X.W.; Writing—original draft, W.C., X.W. and B.Y.; Writing—review and editing, J.C., T.J. and J.S. All authors have read and agreed to the published version of the manuscript.

**Funding:** This research was funded by the National Natural Science Young Foundation of China under Grant 41806117; the Marine Science and Technology Innovation Project of Jiangsu Province under Grant JSZRHYKJ202201; the National Natural Science Foundation of China under Grant 41004003; the Science and Technology Department Project of Jiangsu Province under Grant BE2016701; the Water Conservancy Science and Technology Project of Jiangsu Province under Grant 2020058 and 2021049; and the Lianyungang 521 Project Research Funding Project under Grant LYG06521202131.

**Data Availability Statement:** Not applicable.

**Conflicts of Interest:** The authors declare no conflict of interest.

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
