# Peer review of "Gas Plume Target Detection in Multibeam Water Column Image Using Deep Residual Aggregation Structure and Attention Mechanism"

_remotesensing, doi:10.3390/rs15112896_

Round 1
Reviewer 1 Report
1. 2. 2.1 - 2.2.4 are all entitled "Input Module". It is suggested that they should be given the representitive title.
2. It is mentioned in the paper that "we add the SimAM... and improve the robustness of the model". Is the robustness mainly represented by the mAP50 and mAP50:95? If so, there is only one category in this paper, the how the robustness can be proved by AP of one category?
3. The meaning of F1 should be explained at line 402.
3
The quality of English is good. The paper is easy to understand.
Reviewer 2 Report
This paper proposes an improved network structure for detecting gas plume targets in water column imaging applications. The idea has novelty and the new network is proposed to accelerate model convergence of the has plume detection when using multibeam images. The paper is well structured and oriented directly to the application of the algorithm. I only suggest checking for typos and grammar mistakes.
I only suggest checking for typos and grammar mistakes.
Reviewer 3 Report
The main question addressed in this paper is: "How can the detection of gas plume targets in water column images (WCIs) be improved?" The paper proposes an enhanced YOLOv7 network structure to achieve more accurate and robust detection of gas plume targets in WCIs.
Moreover, although the proposed network structure has exhibited promising outcomes, it is imperative to investigate its generalizability across diverse datasets and environmental conditions. Conducting extensive testing on a larger and more varied dataset will enable a thorough evaluation of its performance, highlighting any potential difficulties encountered in real-world scenarios.
Although the article carries significance within the field, it would benefit from a more explicit and comprehensive explanation of its novelty and distinctive contributions to further distinguish it from existing research. This would help readers better appreciate the unique aspects and advancements presented in the article.
The paper would benefit from a more thorough analysis and comparison to elucidate the specific advantages of the proposed method over established state-of-the-art approaches like Faster R-CNN, SSD, RetinaNet, Mask R-CNN, and EfficientDet. This could involve conducting quantitative evaluations, performance metrics, and benchmarking experiments to demonstrate superior accuracy, speed, robustness, or other desirable characteristics.
In the conclusion section, it is pertinent to incorporate a discussion of the limitations encountered during the research and propose potential future work aimed at addressing these limitations.
Round 2
Reviewer 3 Report
No further comments, accept.